# Prevalence and risk factors of lower extremity disease in high risk groups in Malawi: a stratified cross-sectional study

Stephen Kasenda ,[1] Amelia Crampin,[1,2] Justine Davies,[3,4,5] Jullita Kenala Malava,[1] Stella Manganizithe,[1] Annie Kumambala,[1] Becky Sandford[6]

For numbered affiliations see end of article.

**Correspondence to**
Professor Justine Davies;
j.davies.6@bham.ac.uk

## ABSTRACT

**Objective** Low/middle-income countries face a disproportionate burden of cardiovascular diseases. However, among cardiovascular diseases, burden of and associations with lower extremity disease (LED) (peripheral arterial disease and/or neuropathy) is neglected. We investigated the prevalence and factors associated with LED among individuals known to have cardiovascular disease risk factors (CVDRFs) in Malawi, a low-income country with a significant prevalence of CVDRFs.

**Design** This was a stratified cross-sectional study.

**Setting** This study was conducted in urban Lilongwe Area 25, and the rural Karonga Health and Demographic Surveillance Site.

**Participants** Participants were at least 18 years old and had been identified to have two or more known CVDRFs.

**Main outcome measures** LED—determined by the presence of one of the following: neuropathy (as assessed by a 10 g monofilament), arterial disease (absent peripheral pulses, claudication as assessed by the Edinburgh claudication questionnaire or Ankle Brachial Pulse Index (ABPI) <0.9), previous amputation or ulceration of the lower limbs.

**Results** There were 806 individuals enrolled into the study. Mean age was 52.5 years; 53.5% of participants were men (n=431) and 56.7% (n=457) were from the rural site. Nearly a quarter (24.1%; 95% CI: 21.2 to 27.2) of the participants had at least one symptom or sign of LED. 12.8% had neuropathy, 6.7% had absent pulses, 10.0% had claudication, 1.9% had ABPI <0.9, 0.9% had an amputation and 1.1% had lower limb ulcers. LED had statistically significant association with increasing age, urban residence and use of indoor fires.

**Conclusions** This study demonstrated that a quarter of individuals with two or more CVDRFs have evidence of LED and 2.4% have an amputation or signs of limb threatening ulceration or amputation. Further epidemiological and health systems research is warranted to prevent LED and limb loss.

## INTRODUCTION

Peripheral vascular disease and peripheral neuropathies (including the spectrum of diabetic foot disease) share a common group of overlapping risk factors, namely cardiovascular disease risk factors (CVDRFs) such as hypertension, diabetes, smoking, obesity and hypercholesterolaemia. Infectious agents, particularly HIV, also contribute to an increased risk of peripheral vascular disease.[1] Resources required to optimise preventative strategies, treat complications and offer rehabilitation after surgery for peripheral vascular disease and peripheral neuropathies disease are largely the same whether the underlying disease process was predominantly ischaemic or neuropathic. As such, we have considered these conditions as one entity representing a spectrum of problems under the term 'lower extremity disease' (LED). Individuals with LED are at risk of lower limb amputation, particularly in cases where there is significant ischaemic necrosis and infection. Amputation can be a life-saving procedure, relieving pain and sepsis, but may also be associated with negative physical, psychological, social and economic consequences.[2,3]

With increasing prevalence of CVDRF in low/middle-income countries (LMICs) superimposed on a background of chronic infections, it is strongly suspected that the associated risk of lower limb amputation will become increasingly important for individuals, families, health systems, and economies.[4] Evidence from LMICs is lacking but suggestive that burden of amputations related to LED conditions will grow. Almost 70% of people with peripheral arterial disease live in LMICs; African countries in particular

## STRENGTHS AND LIMITATIONS OF THIS STUDY

⇒ This study is the first attempt in Malawi to determine the prevalence of lower extremity disease (LED).
⇒ This study used multiple parameters to ensure accurate estimation of the prevalence of LED.
⇒ This study included both rural and urban populations to provide a comprehensive picture of LED prevalence.
⇒ Some participants with vascular calcification may have had false negative Ankle Brachial Pulse Index results.

show a high population-prevalence of peripheral arterial disease (e.g., 33% in a population of people >65 years old in Central Africa).[5–7] Studies in people who have managed to access CVDRF clinics in referral hospitals also show that a large number of people have peripheral artery disease; for example, in Ghana, 27% of nearly 1000 people with diabetes had clinically measurable peripheral arterial disease.[8] In our pilot study of 191 patients attending a hospital clinic in Blantyre, Malawi, peripheral arterial disease was seen in 8.5% of 45–64 year olds and 17% of people aged 65 and over.[9] Reliable data on population prevalence of diabetic foot ulceration are lacking, however, studies show that in people with diabetes who have accessed services, the prevalence ranged from 4.0% to 9.9%; it could be far higher if people who had not accessed services were included.[10]

To develop health systems to manage and prevent LED and its consequences requires knowledge of its prevalence. The aim of this study was to investigate the population prevalence of LED among high risk individuals in rural and urban Malawi.

## METHODS

### Study setting

The study was conducted in Area 25 in urban Lilongwe, and the rural Karonga Health and Demographic Surveillance Site (HDSS). The Karonga site has a population of approximately 45 000, and the economy is mainly subsistence farming. Area 25 (Lilongwe) has an approximate total population of 65 000 with a mixed economy.

### Study design and participants

We conducted a stratified cross-sectional study of the prevalence of LED, defined as neuropathy and/or vasculopathy, in all adults of at least 18 years age with two or more CVDRFs. This population was selected to ensure a reasonable sample size of people at the highest risk of LED.

### Identification of participants

We included all eligible participants who were at least 18 years old and had been identified to have two or more known risk factors for LED during a population wide survey of cardiometabolic conditions conducted between 2013 and 2017 in the study areas.[11] Presence of hypertension, diabetes mellitus, HIV, tobacco smoking, obesity, and age at least 40 years were used to characterise high risk individuals.

### Data collection and definitions

Data on participant sociodemographic characteristics (sex, date of birth, marital status, highest attained education, use of indoor fire, and household wealth) and presence of CVDRFs (history of smoking, diabetes mellitus, hypertension, dyslipidaemia, and HIV status using previous laboratory diagnosis) were extracted from the initial baseline study. Methods used to capture data

and definitions used in that baseline survey have been published elsewhere.[11] In brief, diabetes mellitus was defined as a fasting blood glucose of at least 7.0 mmol/L (as determined by a Beckman Coulter AU480 Chemistry Analyser) or self-report of a previous diagnosis of the condition by a health professional regardless of the drug history. Hypertension was defined as a blood pressure of at least 140/90 mm Hg or current use of antihypertensive medication for blood pressure control. Participants with a body mass index of at least 30 kg/m$^2$ were classified as having obesity. Participants were defined as smokers if they reported smoking at least one cigarette per day in the immediate past 6 months preceding the survey. Participants were defined as HIV-positive if they had previously been tested antibody positive by the study team or self-reported HIV positivity.

For the current study, participants were interviewed and examined in their own homes, and data collectors were blind to the participant's previous medical history. In cases where the potential participant was missed during the first visit, and it was established that they had neither left the study area nor died, the household was visited at least three times before declaring the potential participant as 'missed'. All participants were interviewed to determine if they had any changes in cardiovascular disease risk profile. The Edinburgh claudication questionnaire was used to determine the presence of claudication and participants who reported the presence of clinical CVDRFs were asked additional questions regarding access to care for the relevant conditions they had.

Participants were examined to determine the presence of leg or foot ulcers, palpable peripheral pulses (brachial, posterior tibial, and dorsalis pedis) or lower limb amputations in both extremities. Neuropen 10 g monofilaments were used to test for the presence of peripheral neuropathy on the plantar surfaces of the great toe aand over the metatarsal heads of the great, 3rd and 5th toes of both feet. Portable sphygmomanometers (Welch Allyn DS55 Durashock hand aneroid model) and handheld continuous wave Doppler ultrasound devices (Bistos Hi.dop BT-200 vascular Doppler with 8 MHz probe) were used to determine the Ankle Brachial Pressure Index (ABPI). The ABPI for each participant was determined according to the methods recommended by Aboyans et al.[12 13] A low ABPI reading suggests peripheral arterial disease, with the blood pressure reduced at the ankle compared with the arm, which implies stenosis within the arterial tree supplying the lower limbs. Each participant was considered to have low ABPI if they had ABPI <0.9 on any or both sides.

### Outcomes

The primary outcome of interest was the presence of any symptom or sign suggestive of LED, termed 'LED prevalence' (history of claudication, limb ulcers, loss of sensation on the plantar surfaces of feet, ABPI <0.9, limb amputation, or absent peripheral pulses). Secondary outcomes were these as individual outcomes.

## Statistical analysis

Given the dearth of information on the prevalence of LED in the local setting, we based our sample size calculation on pilot data from a study of 90 people with diabetes living in the Malawi HDSS, which suggested that the prevalence of LED was around 10%. Considering people with diabetes to be at high risk and similar to those chosen for our study, we calculated a sample of 912 people would be required to detect a prevalence of LED of 10% with a CI of 6% to 14%.

Prevalence of the primary and secondary outcomes, sociodemographic characteristics of the study participants and CVDRFs are described as mean (SD) and count (%). Logistic regression and likelihood ratio tests were done to determine associations between the primary outcome and independent variables. Sociodemographic characteristics and the presence of CVDRFs were considered as a priori independent variables of interest in our analysis. The association between the number of CVDRFs and the crude prevalence of the outcome of interest was also investigated.

All variables of interest were added to the regression model, and likelihood ratio testing was done to determine the presence of interactions between the independent variables. Stratified analyses of the full regression model were done for all variables that were identified as having statistically significant interactions. All statistical analysis was done using Stata V.15 statistical package.

### Role of the funding source

The funders had no role in the study design; collection, analysis and interpretation of data; or report writing. The corresponding author had full access to the data and the final responsibility to submit for publication.

### Patient and public involvement

There was no patient and public involvement in this study.

## RESULTS

A total of 829 people were visited, of whom 806 (97.2%) consented to be recruited (figure 1). Most of the participants were at least 50 years old (mean age 52.5; 95% CI: 51.7 to 53.3), male (53.5%; n=431), from the rural site (56.7%; n=457), employed (70.6%; n=569), married (74.1%; n=597) and had no more than primary education (61.1%; n=493). Cooking with biomass fuel was reported by 94.8% (n=764) of the participants (table 1). The majority, 88.1% (n=710), of participants, had at least two clinical risk factors for LED (hypertension, obesity, HIV, diabetes, and dyslipidaemia), and the remaining 96 participants were aged at least 40 years and had a history of smoking tobacco (figure 2 and table 1). The mean age of the rural participants was higher than that of the urban participants (53.5 years (95% CI: 52.4 to 54.6) compared with 51.2 years (95% CI: 50.1 to 52.3); p=0.005). Missing data accounted for not more than 6.1% of all the data on any variable.

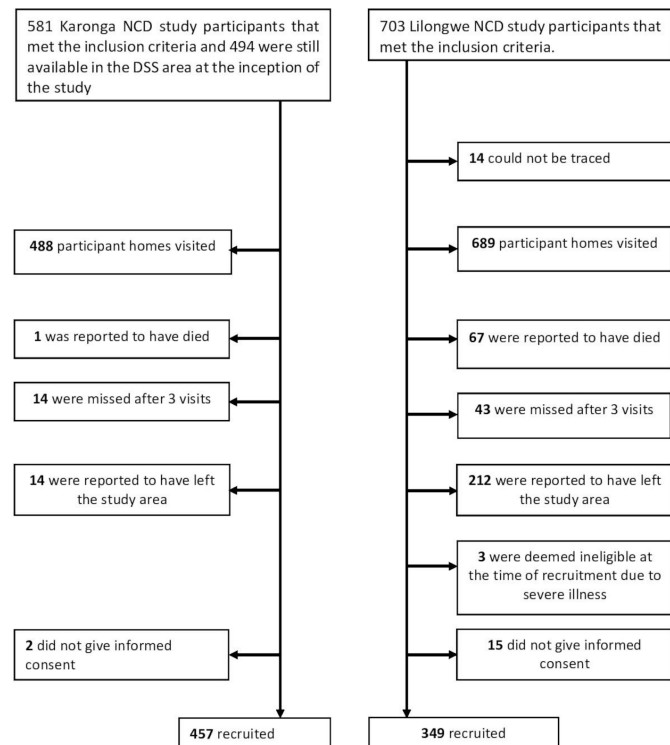

**Figure 1** Flowcharts of study participation by site. DSS, Demographic Surveillance Site; NCD, Non-Communicable Diseases.

Nearly a quarter (24.1%; 95% CI: 21.2 to 27.2) of the participants had at least one symptom or sign of LED (table 2). Peripheral neuropathy (n=103; 12.8%) and claudication (n=81; 10%) were the most common features of LED, and signs of limb threatening disease (ie, ulcers (n=13; 1.1%) and lower limb amputations (n=7; 0.9%)) were the least prevalent, affecting 2.4% (n=19) of participants. ABPI <0.9 was recorded among 15 (1.9%) participants, and 391 (48.5%) had ABPI >1.2. (online supplemental table 1). Four participants had ABPI <0.9 on one side and ABPI >1.2 on the contralateral side. In view of the reduced ABPI <0.9, which is more likely to be a true positive, these individuals were categorised as having low ABPI. Those with ABPI >1.2 on both sides were grouped with participants with ABPI between 0.9 and 1.2. At least 25% of participants with each CVDRF, apart from smoking, had LED. LED prevalence rates of over 30% were recorded among participants who were urban residents, women, aged over 50 years, single, indoor users of biomass fuel or homemakers. The prevalence of LED was directly proportional to the number of CVDRFs with the highest prevalence of LED (35.6%; 95% CI: 23.6 to 49.1) being recorded among participants that had five or more risk factors (table 2).

In the full regression model (figure 3), statistically significantly increased odds of LED were found in participants aged at least 50 years ((OR: 1.19; 95% CI: 1.22 to 2.99) among those aged 50–60, and (OR: 2.33; 95% CI: 1.43 to 3.8) among those over 60 years old) and urban residents (OR: 1.76; 95% CI: 1.05 to 2.94) (online

**Table 1** Summary of participant characteristics

| Variables | n (%) | Both sites | | Karonga (rural) | | Lilongwe (urban) | |
|---|---|---|---|---|---|---|---|
| | | Men | Women | Men | Women | Men | Women |
| Total | | **431** | **375** | **303** | **154** | **128** | **221** |
| Age | | | | | | | |
| Under 50 | 346 (42.9) | 199 (46.2) | 147 (39.2) | 138 (45.5) | 51 (33.1) | 61 (47.7) | 96 (43.4) |
| 50–60 | 239 (29.7) | 113 (26.2) | 126 (33.6) | 80 (26.4) | 49 (31.8) | 33 (25.8) | 77 (34.8) |
| Over 60 | 221 (27.4) | 119 (27.6) | 102 (27.2) | 85 (28.1) | 54 (35.1) | 34 (26.6) | 48 (21.7) |
| Wealth quintiles | | | | | | | |
| Poorest | 108 (13.4) | 41 (9.5) | 67 (17.9) | 34 (11.2) | 51 (33.1) | 7 (5.5) | 16 (7.2) |
| Second | 150 (18.6) | 83 (19.3) | 67 (17.9) | 80 (26.4) | 46 (29.9) | 3 (2.3) | 21 (9.5) |
| Third | 139 (17.3) | 85 (19.7) | 54 (14.4) | 77 (25.4) | 29 (18.8) | 8 (6.3) | 25 (11.3) |
| Fourth | 191 (23.7) | 103 (23.9) | 88 (23.5) | 66 (21.8) | 18 (11.7) | 37 (28.9) | 70 (31.7) |
| Wealthiest | 218 (27.1) | 119 (27.6) | 99 (26.4) | 46 (15.2) | 10 (6.5) | 73 (57) | 89 (40.3) |
| Marital status | | | | | | | |
| Single | 209 (25.9) | 54 (12.5) | 155 (41.3) | 36 (11.9) | 82 (53.3) | 18 (14.1) | 73 (33.0) |
| Married | 597 (74.1) | 377 (87.5) | 220 (58.7) | 267 (88.1) | 72 (46.8) | 110 (85.9) | 148 (67.0) |
| Highest attained education | | | | | | | |
| 0–5 years primary education | 164 (20.4) | 59 (13.7) | 105 (28) | 48 (15.8) | 66 (42.9) | 11 (8.6) | 39 (17.7) |
| Standard 6–8 | 329 (40.8) | 184 (42.7) | 145 (38.7) | 156 (51.5) | 65 (42.2) | 28 (21.9) | 80 (36.2) |
| Secondary | 230 (28.5) | 141 (32.7) | 89 (23.7) | 94 (31.0) | 21 (13.6) | 47 (36.7) | 68 (30.8) |
| Post-secondary | 83 (10.3) | 47 (10.9) | 36 (9.6) | 5 (1.7) | 2 (1.3) | 42 (32.8) | 34 (15.4) |
| Occupation | | | | | | | |
| Homemaker | 237 (29.4) | 74 (17.2) | 163 (43.5) | 32 (10.6) | 34 (22.1) | 42 (32.8) | 129 (58.4) |
| Farming/fishing | 296 (36.7) | 209 (48.5) | 87 (23.2) | 208 (68.7) | 84 (54.6) | 1 (0.8) | 3 (1.36) |
| Employed | 273 (33.9) | 148 (34.3) | 125 (33.3) | 63 (20.8) | 36 (23.4) | 85 (66.4) | 89 (40.3) |
| Indoor fire | | | | | | | |
| No | 42 (5.2) | 26 (6.0) | 16 (4.3) | 17 (5.6) | 6 (3.9) | 9 (7.0) | 10 (4.5) |
| Yes, inside the house | 39 (4.8) | 19 (4.4) | 20 (5.3) | 5 (1.7) | 3 (2.0) | 14 (10.9) | 17 (7.7) |
| Yes, but in a separate kitchen | 725 (90.0) | 386 (89.6) | 339 (90.4) | 281 (92.7) | 145 (94.2) | 105 (82) | 194 (87.8) |
| Smoking history | | | | | | | |
| Non-smokers | 487 (60.4) | 130 (30.2) | 357 (95.2) | 72 (23.8) | 142 (92.2) | 58 (45.3) | 215 (97.3) |
| Smokers | 319 (39.6) | 301 (69.8) | 18 (4.8) | 231 (76.2) | 12 (7.8) | 70 (54.7) | 6 (2.7) |
| Diabetes | | | | | | | |
| Non-diabetic | 502 (62.3) | 324 (75.2) | 324 (75.2) | 251 (82.8) | 88 (57.1) | 73 (57) | 90 (40.7) |
| Diabetic | 273 (33.9) | 87 (20.2) | 87 (20.2) | 42 (13.9) | 62 (40.3) | 45 (35.2) | 124 (56.1) |
| Missing data | 31 (3.9) | 20 (4.6) | 20 (4.6) | 10 (3.3) | 4 (2.6) | 10 (7.8) | 7 (3.2) |
| Dyslipidaemia | | | | | | | |
| Yes | 344 (42.7) | 238 (55.2) | 175 (46.7) | 177 (58.4) | 73 (47.4) | 61 (47.7) | 102 (46.2) |
| No | 413 (51.2) | 165 (38.3) | 179 (47.7) | 114 (37.6) | 74 (48.1) | 51 (39.8) | 105 (47.5) |
| Missing data | 49 (6.1) | 28 (6.5) | 21 (5.6) | 12 (4.0) | 7 (4.6) | 16 (12.5) | 14 (6.3) |
| Hypertension | | | | | | | |
| Hypertensive | 482 (59.8) | 220 (51.0) | 104 (27.7) | 171 (56.4) | 40 (26.0) | 49 (38.3) | 64 (29.0) |
| Not hypertensive | 324 (40.2) | 211 (49.0) | 271 (72.3) | 132 (43.6) | 114 (74.0) | 79 (61.7) | 157 (71.0) |
| Mean blood pressure (SD) | 139.1 (38.2) | 136 (41.6) | 142.6 (33.7) | 137 (47) | 149.1 (43.6) | 133.7 (24.1) | 138.1 (23.6) |
| HIV | | | | | | | |
| Negative | 413 (51.2) | 237 (55.0) | 176 (46.9) | 170 (56.1) | 80 (52.0) | 67 (52.3) | 96 (43.4) |

**Table 1** Continued

| Variables | n (%) | Both sites | | Karonga (rural) | | Lilongwe (urban) | |
|---|---|---|---|---|---|---|---|
| | | Men | Women | Men | Women | Men | Women |
| Positive | 247 (30.7) | 107 (24.8) | 140 (37.3) | 78 (25.7) | 57 (37.0) | 29 (22.7) | 83 (37.6) |
| Unknown | 146 (18.1) | 87 (20.2) | 59 (15.7) | 55 (18.2) | 17 (11.0) | 32 (25) | 42 (19) |
| Obesity | | | | | | | |
| No | 634 (78.7) | 402 (93.3) | 232 (61.9) | 292 (96.4) | 113 (73.4) | 110 (85.9) | 119 (53.9) |
| Yes | 172 (21.3) | 29 (6.7) | 143 (38.1) | 11 (3.6) | 41 (26.6) | 18 (14.1) | 102 (46.2) |

supplemental table 2). Cooking with methods other than biomass burning was negatively associated with LED (OR: 0.19; 95% CI: 0.05 to 0.69). The following sets of variables had some statistically significant interactions that affected the odds of the LED: (1) sex and history of smoking, (2) obesity and highest attained education and (3) obesity and participant's wealth quintile, as computed in the full model. The results of these interactions are presented in online supplemental tables 3–6.

## DISCUSSION

This study aimed to describe the prevalence of LED among individuals with known CVDRFs in rural and urban Malawi. Of the 806 individuals studied, a large proportion had at least one sign of LED, with claudication and peripheral neuropathy being the most common. The prevalence of LED was directly related to the number of risk factors with advancing age, urban living, and poverty demonstrating the greatest association. Although amputations and active ulceration were less frequently seen, 2.4% of individuals in this study had these conditions which are indicative of active or previous limb threatening disease.

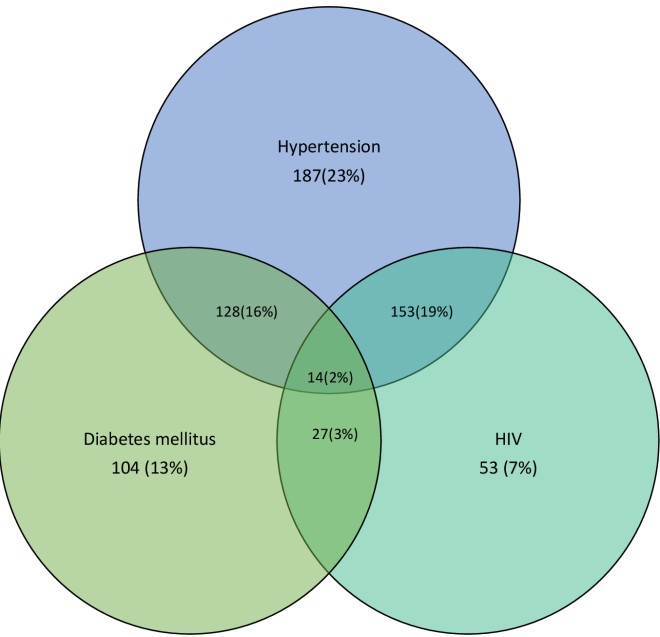

**Figure 2** Three major risk factors and multimorbidity across the study population.

Provision of healthcare services to manage CVDRFs or care for patients with LED is limited in Malawi and tends to be focused around large urban centres.[14] Health literacy also tends to be poor among those living with chronic conditions, with patients and families often understanding little about their condition or the longer term consequences of inadequate management.[15] Provision of surgical services for the management of LED (eg, vascular reconstructive techniques, amputation, and rehabilitation services) is limited throughout southern Africa, and this field has not been viewed as a priority among competing healthcare development needs.[16 17]

In poorly resourced health systems where people present late with disease, amputation rates and subsequent mortality is high. For example, a small study in Tanzania showed that 33% of people admitted to hospital with diabetic foot disease were managed by amputation and mortality was nearly 50%.[18] By comparison, in the UK 17% of people presenting with disease underwent amputation within 1 year of presentation with no reported mortality.[19] After limb loss, poor postoperative care and the dearth of services for rehabilitation, prostheses, and mobility aids result in further mortality and morbidity.[20] The devastating effects are tragic given there are simple and cheap strategies—including preventing and treating CVDRFs and regular monitoring for signs of LED in high risk individuals with early referral to specialist services—to reduce deleterious outcomes.

Although this is one of the few studies to look at the population prevalence of LED,[7] the findings of this study are in keeping with previously published work.[8 21] For example, population prevalence of vascular disease has been reported at over 30% in those over 65 years.[7]

An interesting 'U' shaped association between LED and wealth was observed, with the greatest prevalence of disease being in the poorest and the wealthiest groups. The prevalence of hypertension and diabetes in these groups was also significantly higher at the extremes of wealth than in the middle three quintiles, which may go some way to explain the observations. There may be other factors, beyond the scope of this study to characterise, such as lack of access to skilled healthcare services for optimum management of these comorbidities in the poorer communities,[22] and potentially other social drivers towards worse ill-health such as sedentary lifestyle and westernised dietary habits among the wealthier group,

**Table 2** Prevalence of led in total and by participant characteristics

| Variables | n | LED prevalence (95% CI) n=194 |
|---|---|---|
| Total | 806 | 24.1 (21.2 to 27.2) |
| Age | | |
| <50 | 346 | 15.9 (12.2 to 20.2) |
| 50–60 | 239 | 27.6 (22.0 to 33.7) |
| >60 | 221 | 33.0 (26.9 to 39.7) |
| Sex | | |
| Male | 431 | 15.8 (12.5 to 19.6) |
| Female | 375 | 33.6 (28.8 to 38.6) |
| Residence | | |
| Rural | 457 | 17.5 (14.1 to 21.3) |
| Urban | 349 | 32.7 (27.8 to 37.9) |
| Wealth quintiles | | |
| Poorest | 108 | 32.4 (16.2 to 30.2) |
| Second | 150 | 22.7 (16.2 to 30.2) |
| Third | 139 | 15.8 (10.2 to 23.0) |
| Fourth | 191 | 24.1 (18.2 to 30.8) |
| Wealthiest | 218 | 26.1 (20.4 to 32.5) |
| Marital status | | |
| Single | 209 | 34.0 (27.6 to 40.8) |
| Married | 587 | 20.6 (17.4 to 24.1) |
| Educational achievement | | |
| 0–5 years primary education | 164 | 25 (18.6 to 32.3) |
| Standard 6–8 | 329 | 25.5 (20.9 to 30.6) |
| Secondary | 230 | 21.7 (16.6 to 27.6) |
| Post-secondary | 83 | 22.9 (14.4 to 33.4) |
| Occupation | | |
| Homemaker | 237 | 37.1 (31.0 to 43.6) |
| Farming/fishing | 296 | 15.5 (11.6 to 20.2) |
| Employed | 273 | 22.0 (17.2 to 27.4) |
| Indoor fire | | |
| None | 42 | 7.1 (1.5 to 19.5) |
| Fire usually lit Inside the house | 39 | 30.8 (17.0 to 47.6) |
| Fire usually lit in a separate kitchen area | 725 | 24.7 (21.6 to 28.0) |
| Any smoking history | | |
| No | 487 | 29.4 (25.4 to 33.6) |
| Yes | 319 | 16.0 (12.1 to 20.5) |
| Diabetes | | |
| Non-diabetic | 502 | 19.7 (16.3 to 23.5) |
| Diabetic | 273 | 33.7 (28.1 to 39.6) |
| Missing data | 31 | 9.7 (2.0 to 25.8) |
| Dyslipidaemia | | |

Continued

**Table 2** Continued

| Variables | n | LED prevalence (95% CI) n=194 |
|---|---|---|
| No | 413 | 22.8 (18.8 to 27.1) |
| Yes | 344 | 26.5 (21.9 to 31.5) |
| Missing data | 49 | 18.4 (8.8 to 32.0) |
| HIV | | |
| Negative | 413 | 23.0 (19.0 to 27.4) |
| Positive | 247 | 23.9 (18.7 to 29.7) |
| Unknown | 146 | 27.4 (20.3 to 35.4) |
| Hypertension | | |
| Not-hypertensive | 324 | 21.9 (17.6 to 26.8) |
| Hypertensive | 482 | 25.5 (21.7 to 29.7) |
| Obesity | | |
| No | 634 | 20.8 (17.7 to 24.2) |
| Yes | 172 | 36.0 (28.9 to 43.7) |
| Number of risk factors | | |
| 2 | 201 | 14.9 (10.3 to 20.6) |
| 3 | 342 | 23.1 (18.7 to 27.9) |
| 4 | 204 | 32.4 (26.0 to 39.2) |
| ≥5 | 59 | 32.2 (20.6 to 45.6) |
| Prevalence of outcomes by risk factors | | |
| No diabetes and no hypertension | 193 | 14.0 (25.6 to 42.4) |
| Diabetes but no hypertension | 131 | 33.6 (25.6 to 42.4) |
| Hypertension but no diabetes | 340 | 22.1 (17.8 to 26.8) |
| Hypertension and diabetes | 142 | 33.8 (26.1 to 42.2) |

LED, lower extremity disease.

which somewhat negate improved access to healthcare for those who are more wealthy.[23–25] It will be important to fully understand these associations in future work to understand which interventions may be the most effective in preventing LED among different population groups.

Smoking status in this study was not shown to be a significant factor associated with LED, in contrast to a well-established body of evidence from high-income countries.[26] The reasons for this are not clear, however, smoking is not as common in Malawi as in some high-income countries and the cut-off for being defined as a smoker in this study was more than six cigarettes per week, which is a relatively low level of consumption.

This was a large-scale population-based study, highlighting those at risk of limb loss among a group with known CVDRF. In focusing on population level data, the study reduces the potential for bias towards those who have already engaged with healthcare services. This is an advantage over much of the published literature to date,

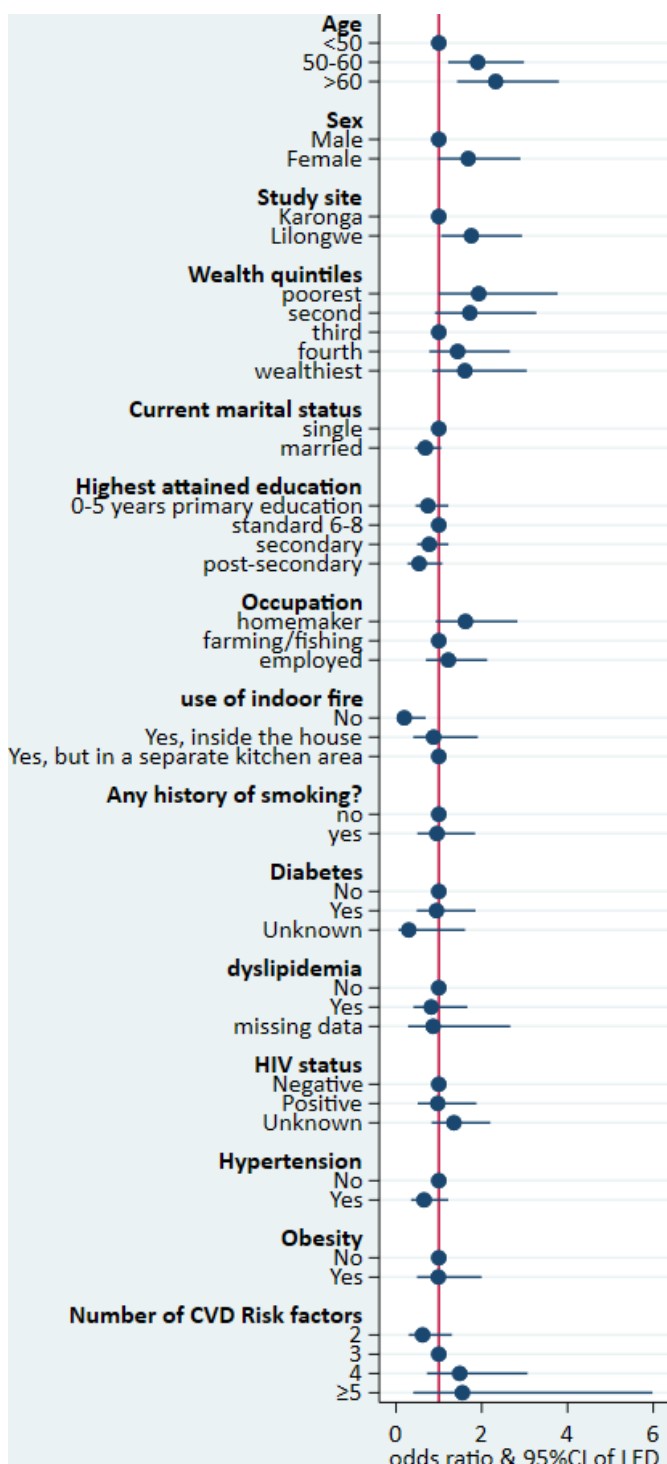

**Figure 3** Coefficient plot of factors associated with lower extremity disease. CVD, cardiovascular disease; LED, lower extremity disease.

and provides a greater understanding of the scale of LED across different population groups in Malawi. The use of multiple approaches to evaluate the presence of LED is also a strength of this study, recognising that no single assessment is either sensitive or specific in isolation.[4]

There are potential limitations to the conclusions to be drawn from this study. The use of ABPI and the Edinburgh Claudication Questionnaire as surrogate markers for peripheral vascular disease, while both validated techniques, do have drawbacks. Claudication, in itself, may never progress to represent a limb-threatening condition, however, can be a good marker of more generalised cardiovascular ill-health and therefore highlight opportunities for optimisation of best medical therapy in order to reduce cardiovascular mortality.[27] ABPI may be confounded by calcification in vessels, resulting in falsely elevated ratios; this is particularly seen among diabetic individuals. The inclusion of those with ABPI >1.2 into the group with ABPI 0.9–1.2 may have resulted in some false negative results, neglecting to report arterial disease which was in fact present. While Toe Brachial Pressure Index (TBPI) may be a more reliable observation in this group,[28] there is little published data on TBPI studies in Sub-Saharan Africa and therefore interpretation of the findings from this study in the context of previously published work would have been more limited had this technique been used. We also did not enquire about the aetiology of ulcers hence it is not possible to establish the aetiology of the ulcers. Finally, we failed to achieve the sample size for this study, however, the proportion of participants affected by LED was far greater than we had estimated and would have required a lower number of participants to reliably detect. In a retrospective calculation, 460 participants would have been required to detect a prevalence of LED of 24% with the same width of CI as in our initial power calculation.

In conclusion, this study demonstrated that 1 in 4 individuals with two or more CVDRFs have evidence of LED and 2.4% already had active signs of limb threatening ulceration or previous amputation. An interesting U-shaped association with wealth was observed. Future work will need to explore the social determinants of health in relation to LED in order to guide health system interventions appropriately; the solutions may be complex and it is likely that there will not be a 'one size fits all' approach to preventing limb loss in this region. It is likely that both therapeutic and preventive services will have to be developed in parallel.

**Author affiliations**
[1]Malawi Epidemiology and Intervention Research Unit, Lilongwe, Malawi
[2]Faculty of Epidemiology and Population Health, London School of Hygiene and Tropical Medicine, London, UK
[3]Centre for Global Surgery, Department of Global Health, Stellenbosch University, Cape Town, South Africa
[4]Institute of Applied Health Research, University of Birmingham, Birmingham, UK
[5]Medical Research Council/Wits University Rural Public Health and Health Transitions Research Unit, Faculty of Health Sciences, School of Public Health, University of the Witwatersrand, Johannesburg, South Africa
[6]Department of Vascular Surgery, Guy's and St Thomas' NHS Foundation Trust, London, UK

**Contributors** SK contributed to data collection, analysis, drafting and revising manuscript. AC contributed to data collection, design, analysis and revising manuscript. JD contributed to design, analysis and revising manuscript. SM contributed to data collection and revising manuscript. AK contributed to data collection and revising manuscript. BS contributed to design, analysis and revising manuscript. AC is the guarantor of the article.

**Funding** This work was supported by King's Health Partners and Development Challenge Fund (MC_PC_16048). The baseline survey was funded by the Wellcome Trust (098610/Z/12/Z and 098610/B/12/A).

**Competing interests** None declared.

**Patient and public involvement** Patients and/or the public were not involved in the design, or conduct, or reporting, or dissemination plans of this research.

**Patient consent for publication** Not applicable.

**Ethics approval** This study involves human participants and was approved by Malawi National Health Sciences Research Committee (NHSRC, protocol number #1072) and London School of Hygiene and Tropical Medicine Ethics Committee (protocol number #6303). Ethical approval for the lower extremity disease study was granted by NHSRC approval number #2090 and King's College London BDM Research Ethics Panel (reference: LRS-17/18-7983). Data were collected from potential participants who gave written informed consent. Additional consent was sought from the participants for utilisation of their previously obtained HIV results in data analysis for this study.

**Provenance and peer review** Not commissioned; externally peer reviewed.

**Data availability statement** Data are available upon reasonable request. Data are available upon reasonable request. All requests must be sent to the Malawi Epidemiology and Intervention Research Unit along with sufficient information, including sound justification, to enable proper evaluation of the request.

**ORCID iD**
Stephen Kasenda http://orcid.org/0000-0001-9208-4122

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
