## [Reviewer comments · BMJ Open]

ARTICLE DETAILS

TITLE (PROVISIONAL)	The prevalence and risk factors of lower extremity disease in high risk groups in Malawi: a stratified cross-sectional study
AUTHORS	Kasenda, Stephen; Crampin, Amelia; Davies, Justine; Malava, Jullita; Manganizithe, Stella; Kumambala, Annie; Sandford, Becky

VERSION 1 – REVIEW

REVIEWER	Mandika Wijeyaratne University of Colombo, Surgery
REVIEW RETURNED	28-Aug-2021

GENERAL COMMENTS	Important study in a low income setting where the burden of lower extremity and CV disease is known to be high but poorly researched. The reviewer provided a marked copy with additional comments. Please contact the publisher for full details.
--

REVIEWER	Mariella Catalano Università degli Studi di Milano, biomedical & clinical science
REVIEW RETURNED	02-Sep-2021

GENERAL COMMENTS	The authors conducted a study of great complexity both for the number of the sample and for the conditions chosen for the evaluation of the subjects (their own homes visits). The quality of this method has been confirmed by the adherence of the study population A-The study design is correct but the methods used to assess LED are confusing as well as the results related to the different conditions. The Authors aim is to evaluate: Peripheral vascular disease, peripheral neuropathies, and diabetic foot disease 1- PAD (peripheral arterial disease)- The evaluation has been done considering: absence of pulse - Edinburgh questionnaire - ABI <0.9 - Amputation. Unclear definition of ulcers (leg or foot) possible inclusion of different disease. -It is not clear whether the different results referred to PAD are all to be summed up or whether there are overlaps (eg absence of pulse and claudication?) -ABI >1.3 (not 0.2) should be selected as abnormal and enclosed in the evaluation -It would be relevant to have defined PAD prevalence in urban and rural areas also comparing to the previous study 2-Neuropaty :by Neuropen 3-Diabetic foot: how has been defined/assessed
---

	-The data collected should be analyses giving a more clear evaluation of the different diseases. B- The paper correctly stressed the relevance of socio-economic factor determining several conditions as well as the need for a highest consideration by the health system of Peripheral arterial disease diabetic foot etc A comment on Introduction (21-31) as well as in Discussion (53...) The first need for patients and for population (everywhere) is prevention and early detection. For this reason a large competent medical (not surgical) staff approaching both patients and population should be considered. Amputations can be avoid with prevention. Therapy (medical and surgical) is relevant for patients already affected by the different diseases enclosed in LED. The two lines should go on in parallel but probably a larger medical education on prevention and diagnosis represents a relevant step Conclusion: Accepted with Major revisions
--	--

REVIEWER	Bernd Salzberger University Hospital Regensburg, Infection Control and Inf. Diseases
REVIEW RETURNED	25-Jan-2022

GENERAL COMMENTS	In the manuscript the "full regression model" is addressed. It is not totally clear, how the authors came to this model. Were all interactions (two or three way) analyzed? Were all factors put in (inclusion) or included/excluded by stepwise LR and if by what margins? This must be stated more clearly in the methods section. I am not concerned about the analysis, the dealing with the stratified analysis in the appendix seems sensible. It is just the information missing in the methods section.
--

REVIEWER	Catherine McCarty University of Minnesota System, Family Medicine and Biobehavioral Health
REVIEW RETURNED	27-Jan-2022

GENERAL COMMENTS	A very interesting article. I have a few questions and suggestions.  1. The authors have used a lot of acronyms, many of which will be unfamiliar to readers. I suggest spelling out many of the less familiar ones, specifically WQ, ABPI, and CVDRF. 2. Where "stratified" is mentioned in the study design, it would be helpful to state what the strata are. 3. Table 1. It would be useful to add the actual values for the wealth quintiles. How is obesity defined? 4. Appendix, Table 1. It would be helpful to highlight the statistically significant results. There are many statistical tests. Did the authors test for multiple comparisons? 5. Appendix, Table 2. I suggest removing missing and unknown from analyses.
---

VERSION 1 – AUTHOR RESPONSE

Reviewer 1

DFD is the result of PVD and/or neuropathy already mentioned. Diabetes mellitus is a key risk factor already included under CVDRFs.

Diabetic foot disease was listed as a separate disease due to the fact that it has other features e.g. Charcot's foot that are not part of the clinical picture of peripheral vascular disease. We have amended the text to make it clearer

DFD is the product of PVD and Neuropathy already stated.

We have deleted "diabetic foot disease"

There is confusion, lack of clarity in what exactly is being studied. This paper is entirely on Foot Disease rather than Lower Extremity Disease. LED would include venous ulceration as well. Furthermore, foot disease in this paper consists of feet with ischemia and/or neuropathy. Diabetes would be the most common underlying cause. Non-diabetics who are heavy tobacco smokers with PVD would account for the rest.

We have studied individuals who have risk factors which predispose to limb loss. This represents a spectrum of disease incorporating diabetes and its complications, both neuropathic, microvascular and macrovascular, but also those with non-diabetic underlying causes including peripheral arterial disease. The umbrella term of 'lower extremity disease' has been defined in the text and is useful to capture individuals at risk, regardless of aetiology. There are limitations associated with this including ulceration from other causes such as venous, tropical or mixed ulceration, however it is beyond the scope of the study to further investigate this. We have added a comment in the text to reflect this limitation.

'And' rather than 'or'

Thank you for the suggestion. We have amended the text accordingly

How exactly was an individual selected from the population?

The details on participant selection are given in the "identification of participants" section. In short, this study was conducted in communities that have well characterized cardiovascular disease risk profiles since 2013. People involved in this study were all living in the study areas who were at least 18 years old and had at least two known risk factors for LED during a previous community survey of cardio-metabolic conditions (that was conducted in the study areas between 2013 and 2017).

This is incomplete. It should read as, '>18 years with two or more cardiovascular risk factors'.

The text has been amended accordingly

***[Regarding inclusion of people at least years age]* Too young to have significant risk factors or neurovascular complications**

Whilst it is unusual to have young people with LED, some young adult residents of the study communities are known to have high risk cardiovascular disease profiles and other risk factors such as HIV. Since our study is the first of its kind in this setting, we felt it prudent to include all at high risk of disease to develop a comprehensive picture of LED that is representative of the entire population of adults at high risk of LED. We also demonstrated a small but significant prevalence of lower extremity disease in the younger age groups in our pilot study (DOI: 10.4314/ecajs.v25i3.2).

***[Regarding inclusion of people with at least two risk factors]* In that case those with diabetes mellitus ONLY would be excluded. Diabetes is special. Diabetes ALONE is more than the rest put together.**

We included all those with 2 or more risk factors (including age >40 years) in order to investigate the population at most risk of LED. Although we accept that diabetes alone is a significant risk factor, in keeping with your earlier point, lower limb complications in this group below the age of 40 are rare unless there is another risk factor present. This strategy allowed us to study the patients most at risk.

There is some contradiction. It was previously stated that the participants were at least 18 years old.

All people aged at least 18 years were considered eligible to participate in the study but a participant's age was only considered a risk factor for LED if the person was over the age of 40. Thus all participants under the age of 40 had two or more cardiovascular disease risk factors other than age. We have made this clearer in the text.

Leg ulceration implies above ankle ulceration. Leg ulcers are venous unless proven otherwise. The focus of this study is foot disease encapsulating PVD and/or neuropathy.

We included all patients who had active ulceration on the lower limb, including the foot. It was beyond the scope of this study to explore the underlying aetiology for the ulceration. We have stated this as a limitation in the discussion.

In both extremities rather than bilaterally.

Thank you for the suggestion. We have amended the text accordingly.

How was a limb categorized if:

- A) pulses were palpable, but ABPI was <0.9.
- B) ABPI >0.9, but pulses were impalpable as it may be with diabetic calcific disease.
- C) History of claudication, but ABPI >0.9.

We have to limit ourselves to objective measures, ABPI <0.9. The exception is diabetes where pulse impalpability alone (ABPI >0.9) could be taken to indicate PVD. Claudication alone (with >0.9 ABPI, palpable pulses) cannot be categorised as PVD. Clinical diagnosis of claudication has poor specificity for PVD. Furthermore, foot ulceration in the case of a negative monofilament test and/or ABPI >0.9 or palpable pulses does not qualify to be classed as LED.

Our primary outcome measures, as stated in the text ("outcomes" section), were presence of any sign or symptom of lower extremity disease, including history of claudication (irrespective of ABPI) or ABPI less than 0.9 (irrespective of pulse status). We acknowledge that calcified diseased vessels in diabetes may be associated with an ABPI >0.9 and for this reason, individuals with an ABPI >1.2 were excluded as it was impossible to confidently verify the presence or absence of arterial disease without additional imaging.

Symptoms and signs may be absent in some with LED and thus are secondary outcomes. Primary outcomes are the ABPI and the monofilament test. Secondary outcomes can be only be studied in those with a positive monofilament test (loss of protective sensation) AND/OR those with a ABPI <0.9. Only diabetics can be considered to have PVD despite an ABPI >0.9. Foot ulceration/amputation in the ABSENCE of neuropathy and PVD is not LED. Traumatic wounds may become chronic due to poor wound care, osteomyelitis or fungal superinfection in the absence of neuropathy or PVD, particularly in those with diabetes ALONE (WITHOUT tobacco, hypertension, hyperlipidemia or BMI >30). In this study where inclusion requires TWO risk factors, such patients will be not counted.

We used simple definitions in this study to – for the first time – get an indication what the prevalence of LED is in a LMIC setting. Whilst we recognize that there are other causes of the signs and symptoms which we aimed to detect in our study, we believe that in our setting, and in the selected population, these are likely to be minimal.

Limb ulcers include venous ulcers which are the most common. The rest are foot ulcers that are commonly neuropathic and/ or ischaemic. Rarely, foot ulcers may be vasculitic or due to other rare causes. Therefore limb ulceration may OVERESTIMATE Lower Extremity Disease defined by the presence of PVD and Or Neuropathy.

Whilst the case definition for this study may overestimate the proportion of people with established peripheral vascular disease, the approach ensures a more comprehensive assessment of people at risk of future peripheral vascular disease and limb loss. Thus the chosen approach enables more precise estimation of the proportion of people that can benefit from primary and secondary prevention of limb loss and vascular intervention. As there were very few people with foot ulcers (<1.1% of the entire study sample) and that some of the participants with ulcers satisfied other definitions of LED used in this study, we are satisfied that these potential over-estimates will not have affected the overall message of the paper.

Outcomes are primarily

1. PVD
 - a. ABPI <0.9,

b. in diabetes, absent ankle pulses even if ABPI is >0.9

2. Monofilament test positive Neuropathy.

Secondary outcomes are clinical outcomes IN THOSE with the primary LED outcome i.e. Claudication, ulceration and amputation. All these clinical outcomes MAY have OTHER (not PVD, not neuropathy) causes such as spinal canal stenosis, musculoskeletal disorders and non-neuropathic injuries. These outcomes DO NOT indicate the existence of a high risk limb.

We selected the primary outcome measures based on validated scoring systems and techniques. We have chosen an inclusive set of primary outcome measures deliberately, to include your suggestions but also include those with active or previous ulceration and amputation, and those with symptoms consistent with claudication (using a validated tool to distinguish from spinal claudication). Many of the individuals with a more subjective symptom or sign (eg claudication) also had objective evidence of LED and therefore focusing on the purely objective measures above would be unlikely to alter the main findings of this study substantially.

Why and How were 829 selected for the study? from a total population of approximately 110,000 at both sites, How many were >18 years? From that number & how many had >2 risk factors? Out of that number:

1. what proportion was selected, and
2. how?

This information should be in the methods section.

The population cohort from which participants were selected comprised of 30575 adults above the age of 18 (16672 in Lilongwe alone) who had participated in the NCD survey. 6886 had >2 risk factors at the time of the baseline population cohort survey but only 1284 were reported to be present within the geographic area of the study. The rest of the selection process is as detailed in figure 1. Thus, we ultimately selected all potential participants who were within the study area at the time of recruitment. We have made this clearer in the methods section.

It appears there are two high risk groups being studied.

1. >18 y with two or more CV risk factors (n=710).
2. >40y tobacco smokers. (n=96)

It is unclear if group 1 had smokers and if group 2 were free of other risk factors i.e. Diabetes, hypertension, lipids etc. One cannot combine two different groups and present stats as for a single cohort as shown in figures and tables.

We consider all the 806 participants of this study as constituting one group with at least two cardiovascular disease risk factors. Each of participants in the two groups mentioned in the question met this criteria and were thus grouped together.

One cannot combine two different groups and present as though they constitute a single group. Each group must be analysed and presented and discussed separately.

All the 806 participants constitute one group of people with at least two cardiovascular disease risk factors that are distributed across two study sites but having comparable socio-demographic characteristics hence they can be analysed as one group. Analysis of the other categorizations can be considered sub-group analyses.

In 2017, the population of Malawi was approx 17 million and includes a large number of health areas. The population of the two districts studied is approximately 110,000. This sample may not represent the population of Malawi.

The two areas that we include in our population based studies are typical of rural and urban communities. The study recruited a relatively large community-based sample consisting of rural and urban participants with characteristics closely approximating the socio-demographic characteristics of the national population. Since this study focused on people with at least two cardiovascular disease risk factors, the results are representative of people at high risk of cardiovascular disease and not the general population per se. Our study although not claiming to be nationally representative, does give a good picture of the population of people at high risk of cardiovascular diseases in Malawi.

806 are made up from two distinct groups that cannot be mixed up and called one high risk group.

The 806 participants, whilst coming from two different sites, were selected using the same criteria and had comparable characteristics. Hence they can be considered as one group. The other group analyses can be considered sub-groups.

Reviewer 2

PAD (peripheral arterial disease)- The evaluation has been done considering: absence of pulse - Edinburgh questionnaire - ABI <0.9 - Amputation. Unclear definition of ulcers (leg or foot) possible inclusion of different disease.

Both leg and foot ulcers were included in this study due to the risk of limb loss in the presence of multiple cardiovascular disease risk factors. Whilst this approach might have led to an overestimation of people with ischemic, infective and neuropathic causes of limb loss, only seven people had lower limb ulcers as the only feature of LED present. Hence the inclusion of both of leg and foot ulcers did not change the overall message of the paper.

It is not clear whether the different results referred to PAD are all to be summed up or whether there are overlaps (eg absence of pulse and claudication?)

A participant was considered to have LED if they had at least one of the conditions of interest. Where the participant had more than one condition, that was considered as one case and this is highlighted in the paper.

ABI >1.3 (not 0.2) should be selected as abnormal and enclosed in the evaluation

All participants with ABPI>1.3 and no feature of LED were not considered as having LED since we did not have the capacity to isolate those with vascular calcification from those with other pathology. We have listed this as one of the limitations of this study

It would be relevant to have defined PAD prevalence in urban and rural areas also comparing to the previous study

This is the first community-based study of its kind in our setting. The baseline survey mentioned in the methods section focused on cardiovascular disease risk profiles but did not look at any of the features of lower extremity disease. The pilot study mentioned in the introduction was done in a referral hospital and is thus not entirely suitable for comparison.

Table 2 presents the rural and urban prevalence of LED as computed in this study [17.5%; 95%CI: 14.1—21.3 vs 32.7%; 95%CI: 27.8—37.9 respectively]

Neuropathy: by Neuropen

As stated in the data collection and definitions section, these are 10g monofilament devices used to assess for loss of pain sensation

Diabetic foot: how has been defined/assessed. The data collected should be analyses giving a more clear evaluation of the different diseases.

We did not define diabetic foot since we sought to identify the presence of specific abnormalities and not to diagnose diseases within the spectrum of lower extremity disease.

A comment on Introduction (21-31) as well as in Discussion (53...)

The first need for patients and for population (everywhere) is prevention and early detection. For this reason a large competent medical (not surgical) staff approaching both patients and population should be considered. Amputations can be avoid with prevention. Therapy (medical and surgical) is relevant for patients already affected by the different diseases enclosed in LED. The two lines should go on in parallel but probably a larger medical education on prevention and diagnosis represents a relevant step

Thank you for the comment, we agree with this assessment, and we hope that future work will focus on the relevant health service interventions to prevent limb loss, as well as caring for those in whom it has become inevitable. We have made some adaptations to the text to reflect this.

Reviewer 3

In the manuscript the "full regression model" is addressed. It is not totally clear, how the authors came to this model. Were all interactions (two or three way) analyzed? Were all factors put in (inclusion) or included/excluded by stepwise LR and if by what margins? This must be stated more clearly in the methods section. I am not concerned about the analysis, the dealing with the stratified analysis in the appendix seems sensible. It is just the information missing in the methods section.

socio-demographic characteristics, types of cardiovascular disease risk factors present, and the number of cardiovascular disease risk factors present were included in the regression model (see the

“statistical analysis” section), The sociodemographic variables of interest were Age, sex, residential area, wealth quintiles, marital status, highest attained education, occupation, and use of indoor fire, and the cardiovascular disease risk factors of interest were smoking, diabetes mellitus, dyslipidemia, HIV, hypertension, and obesity. All the variables were added to the regression model at once and stratified analyses were done on all variables that had statistically significant interactions. We have amended the text to make these facts clearer.

Reviewer 4

The authors have used a lot of acronyms, many of which will be unfamiliar to readers. I suggest spelling out many of the less familiar ones, specifically WQ, ABPI, and CVDRF.

Thank you for the suggestion. Each acronym has been spelled out at the point of first occurrence within the text. We have removed “WQ” wherever possible and where the abbreviation could not be removed, we have added the meaning of the abbreviation as a footnote on the relevant page.

Where "stratified" is mentioned in the study design, it would be helpful to state what the strata are.

Given the number of stratified analyses done in this study and the number of strata in each analysis, detailing the strata in the manuscript would make the manuscript unnecessarily long. We have submitted additional tables in the appendix (appendix tables 3 to 6), which detail the results of the stratified analyses. The titles of each column in these tables (except the column on the extreme left) indicate the strata of each analysis, hoping this is helpful

Table 1. It would be useful to add the actual values for the wealth quintiles. How is obesity defined?

In a setting where many people have informal occupations, wealth quintiles are determined using socioeconomic status questionnaires, which focus on household assets than monetary household income.

Obesity was defined a body mass index of at least 30Kg/m². See line 14 of the “data collection and definitions” section,

Appendix, Table 1. It would be helpful to highlight the statistically significant results. There are many statistical tests. Did the authors test for multiple comparisons?

All statistically significant results have been highlighted in the main text. We did/did not do tests for multiple comparisons, however the key associations were highly significant. We have commented in the discussion.

Appendix, Table 2. I suggest removing missing and unknown from analyses.

We have left the “missing” & “unknown” descriptions in the tables since leaving the cells blank would necessitate extensive explanation of the reasons behind the blank cells, however they are not included in the calculations.

VERSION 2 – REVIEW

REVIEWER	Catherine McCarty University of Minnesota System, Family Medicine and Biobehavioral Health
REVIEW RETURNED	07-Mar-2022
GENERAL COMMENTS	All of my concerns have been addressed and the authors have addressed concerns raised by the other reviewers.